# Manganese- and Platinum-Driven Oxidative and Nitrosative Stress in Oxaliplatin-Associated CIPN with Special Reference to Ca_4_Mn(DPDP)_5_, MnDPDP and DPDP

**DOI:** 10.3390/ijms25084347

**Published:** 2024-04-15

**Authors:** Jan Olof G. Karlsson, Per Jynge

**Affiliations:** 1Department of Biomedical and Clinical Sciences, Division of Clinical Chemistry and Pharmacology, Linköping University, 581 83 Linköping, Sweden; 2Department of Radiology, Innlandet Trust Hospital, Gjøvik Hospital, 2819 Gjøvik, Norway; per.jynge.ha@gmail.com

**Keywords:** calmangafodipir, mangafodipir, manganese, platinum, MnSOD mimetics, oxaliplatin-associated CIPN, oxidative stress, nitrosative stress, chelation therapy

## Abstract

Platinum-containing chemotherapeutic drugs are efficacious in many forms of cancer but are dose-restricted by serious side effects, of which peripheral neuropathy induced by oxidative–nitrosative-stress-mediated chain reactions is most disturbing. Recently, hope has been raised regarding the catalytic antioxidants mangafodipir (MnDPDP) and calmangafodipir [Ca_4_Mn(DPDP)_5_; PledOx^®^], which by mimicking mitochondrial manganese superoxide dismutase (MnSOD) may be expected to overcome oxaliplatin-associated chemotherapy-induced peripheral neuropathy (CIPN). Unfortunately, two recent phase III studies (POLAR A and M trials) applying Ca_4_Mn(DPDP)_5_ in colorectal cancer (CRC) patients receiving multiple cycles of FOLFOX6 (5-FU + oxaliplatin) failed to demonstrate efficacy. Instead of an anticipated 50% reduction in the incidence of CIPN in patients co-treated with Ca_4_Mn(DPDP)_5_, a statistically significant increase of about 50% was seen. The current article deals with confusing differences between early and positive findings with MnDPDP in comparison to the recent findings with Ca_4_Mn(DPDP)_5_. The POLAR failure may also reveal important mechanisms behind oxaliplatin-associated CIPN itself. Thus, exacerbated neurotoxicity in patients receiving Ca_4_Mn(DPDP)_5_ may be explained by redox interactions between Pt^2+^ and Mn^2+^ and subtle oxidative–nitrosative chain reactions. In peripheral sensory nerves, Pt^2+^ presumably leads to oxidation of the Mn^2+^ from Ca_4_Mn(DPDP)_5_ as well as from Mn^2+^ in MnSOD and other endogenous sources. Thereafter, Mn^3+^ may be oxidized by peroxynitrite (ONOO^−^) into Mn^4+^, which drives site-specific nitration of tyrosine (Tyr) 34 in the MnSOD enzyme. Conformational changes of MnSOD then lead to the closure of the superoxide (O_2_^•−^) access channel. A similar metal-driven nitration of Tyr74 in cytochrome c will cause an irreversible disruption of electron transport. Altogether, these events may uncover important steps in the mechanism behind Pt^2+^-associated CIPN. There is little doubt that the efficacy of MnDPDP and its therapeutic improved counterpart Ca_4_Mn(DPDP)_5_ mainly depends on their MnSOD-mimetic activity when it comes to their potential use as rescue medicines during, e.g., acute myocardial infarction. However, pharmacokinetic considerations suggest that the efficacy of MnDPDP on Pt^2+^-associated neurotoxicity depends on another action of this drug. Electron paramagnetic resonance (EPR) studies have demonstrated that Pt^2+^ outcompetes Mn^2+^ and endogenous Zn^2+^ in binding to fodipir (DPDP), hence suggesting that the previously reported protective efficacy of MnDPDP against CIPN is a result of chelation and elimination of Pt^2+^ by DPDP, which in turn suggests that Mn^2+^ is unnecessary for efficacy when it comes to oxaliplatin-associated CIPN.

## 1. Background

The addition of oxaliplatin to 5-fluorouracil (5-FU) or capecitabine chemotherapy of colorectal cancer according to the FOLFOX or CAPOX regimen has increased tumoricidal efficacy, both in the adjuvant and palliative settings [1,2,3,4,5,6,7,8,9]. However, this is an addition that has increased unwanted toxicity, where chemotherapy-induced peripheral neuropathy (CIPN) is the most problematic dose-limiting toxicity of oxaliplatin. Neutropenia is another troublesome toxicity of oxaliplatin plus 5-FU experienced by many patients. Various approaches to prevent CIPN caused by oxaliplatin and other platinum-containing drugs have generally failed [10]. The exact mechanism behind oxaliplatin- and cisplatin-associated CIPN is poorly understood. However, it seems reasonable to assume that it is the uptake and retention of Pt^2+^ in the peripheral sensory nerve system that cause the toxicity. This system lacks a blood–brain barrier, a draining lymph system, and cerebrospinal fluid [11]. This makes potentially dangerous substances, such as chemotherapy drugs, to accumulate in the peripheral nerve system and cause oxidative stress and detrimental nerve injuries.

Although there are many similarities between the pharmacodynamic, pharmacokinetic, and (unwanted) toxicological properties of oxaliplatin and other platinum-containing drugs, particularly those of cisplatin, there are also major differences. In the current article, we will base our analysis on oxaliplatin with the aim of scrutinizing the possibility of preventing oxaliplatin-associated CIPN by chelation therapy with fodipir (DPDP), similar to that lately obtained with sodium thiosulfate (STS) in pediatric patients (see below).

At the start of oxaliplatin- or cisplatin-based treatment, the terminal elimination half-life of Pt^2+^ is about one month. However, over time, the half-life will increase and exceed 1 year at 10 to 20 years after chemotherapy [12], clearly illustrating the problem with platinum retention. As in other forms of heavy metal toxicity, chelation therapy, i.e., administration of a suitable low-weight metal-binding ligand forming a nontoxic metal complex, small enough for being filtered through the kidneys (or alternatively redistributed to a less critical tissue), appears as an attractive possibility [13].

Chelation therapy to prevent Pt^2+^-associated CIPN has until now remained an unproven option. However, in 2022, the FDA approved co-treatment with the metal chelator sodium thiosulfate (STS) (Pedmark^®^) to prevent severe ototoxicity of cisplatin in the treatment of pediatric patients with localized, non-metastatic solid tumors. The decision was based on two multicenter trials, i.e., the SIOPEL 6 trial and the ACCL0431 trial [14,15]. Importantly, the ACCL0431 design included multiple cancer types at any stage, and post hoc analyses showed that although survival in the STS group was equivalent to that in the control group among patients with localized disease, it was significantly lower among those with disseminated disease [15]. These results reflect the difficulty of finding or creating selective protectants, i.e., compounds that protect against “unwanted” toxicity but not against the “wanted” tumoricidal toxicity. Although chelation therapy obviously does not provide any quick fix, it provides hope that chelation therapy may be a passable way of solving the oxaliplatin-associated CIPN problem.

Multiple lines of evidence indicate that the platinum-containing cancer drugs enter cells, are distributed to various subcellular compartments, and are exported from cells via transporters that evolved to manage copper homeostasis [16] (Figure 1). Copper (Cu^2+^) and Pt^2+^ have similar sulfur-binding characteristics, and the presence of stacked rings of methionines and cysteines in Copper Transporter 1 (CTR1) suggests a mechanism where CTR1 selectively transports copper and platinum-containing drugs via sequential trans-chelation reactions. The similarities in binding characteristics of these metal cations also include nitrogen ligands, where cisplatin- or oxaliplatin-associated Pt^2+^ binds to purine bases, preferentially guanine, which in fact reveal the tumoricidal mechanisms of both oxaliplatin and cisplatin [17].

Interestingly, copper deficiency causes neurological symptoms similar to those seen in Pt^2+^-associated CIPN, and hyper-physiological intake of Zn^2+^ is a common cause of copper deficiency in humans [18]. This may in turn suggest that Pt^2+^ interferes with copper handling in a manner that causes a deficiency and a subsequent increase in oxidative/nitrosative stress in the dorsal root ganglion (DRG) cells. Cytosolic copper superoxide dismutase (CuSOD) and cytochrome c activities depend fully on the presence of redox-active copper. Copper deficiency by itself hence results in oxidative/nitrosative stress causing severe cellular injuries, including DRG cells.

## 2. Randomized Clinical Phase III Trials Failed to Demonstrate Positive Effects of Ca_4_Mn(DPDP)_5_

At the end of 2022, Pfeiffer et al. reported results from the POLAR A and M phase III trials in CRC patients testing the efficacy of the metal complex and MnSOD mimetic Ca_4_Mn(DPDP)_5_ (PledOx^®^) against oxaliplatin-associated CIPN, with the somewhat confusing aim of lowering the incidence of persistent CIPN from 40% to 20% [19]. However, in the preceding PLIANT phase II trial in CRC patients, the patients displayed an exceedingly lower frequency of adverse events, also including CIPN, than those expected from 5-FU plus oxaliplatin [20,21]. The frequency was more in line with what one expects from 5-FU alone.

The POLAR trials failed to show positive efficacy. Instead of a hypothesized 50% improvement in the incidence of persistent CIPN, the real outcome was an about 50% worsening of this highly handicapping toxicity. Mechanisms that may explain the outcome, with a statistically significant number of patients being seriously injured after having received Ca_4_Mn(DPDP)_5_, indicate interactions between Pt^2+^-containing oxaliplatin and Mn^2+^-containing Ca_4_Mn(DPDP)_5_ [22] (Figure 2). The POLAR failures showed with no doubt that the positive effects of Ca_4_Mn(DPDP)_5_ on CIPN claimed by the authors of the PLIANT trial report [20] were not real, in line with Karlsson and Jynge’s criticism [21].

## 3. Promising Clinical Findings with MnDPDP

In 2009, Yri et al. [23] published a case report describing a young patient who received fifteen palliative cycles of oxaliplatin plus 5-FU (“Nordic FLOX” regimen), suggesting that MnDPDP may protect against oxaliplatin-associated CIPN. In fourteen of the cycles, the patient received pretreatment with MnDPDP. The patient received an accumulated dose of 1275 mg/m^2^ oxaliplatin, which is a dose likely to give severe neurotoxic symptoms. However, CIPN was not seen, except for in the fifth cycle, when MnDPDP was left out and the patient experienced CIPN. After five cycles, the performance status of the patient was drastically improved, and the demand for analgesics was reduced. Furthermore, neutropenia did not occur during any of the chemotherapy cycles.

The case report was followed by an important paper by Coriat et al. 2014 [24], published in the *Journal of Clinical Investigation*. The results suggested that co-treatment with MnDPDP (Teslascan™) not only protected the patients from CIPN, but in fact also cured ongoing oxaliplatin-related CIPN. The same group headed by Batteux had previously presented results showing that MnDPDP increased the therapeutic index of both oxaliplatin and paclitaxel by simultaneously increasing the tumoricidal and the cytoprotective activity of these cytostatic drugs [25,26,27]. Karlsson et al., 2012, have reported similar results with Ca_4_Mn(DPDP)_5_ [28]. These findings clearly suggested that MnDPDP or the MnDPDP metabolite Mn pyridoxyl ethyldiamine (MnPLED) might solve the oxaliplatin-associated CIPN problem.

Coriat, Batteux, et al. [24] naturally assumed that the therapeutic effects of MnDPDP were due to its MnSOD-mimetic activity, i.e., a property of lowering oxidative and nitrosative cellular stress by targeting superoxide (O_2_^•−^) (Figure 3). However, this explanation is questionable from a pharmacokinetic perspective, where the MnSOD-mimetic activity lasts only a couple of hours [13,29]. A more plausible explanation is that DPDP or its metabolite PLED acting by chelation of Pt^2+^.

## 4. Confusing Pharmacodynamic Difference between Ca_4_Mn(DPDP)_5_ and MnDPDP

There are few reasons to believe that the difference between Ca_4_Mn(DPDP)_5_ and MnDPDP can be explained by a fundamental pharmacodynamic difference between them. However, one cannot fully exclude such a difference, taking into consideration the complex in vivo metabolism of MnDPDP [29] and the fact that the ready-to-use MnDPDP (Teslascan™) but not Ca_4_Mn(DPDP)_5_ contains ascorbic acid. To fully exclude such a difference, preclinical studies similar to those presented by Coriat et al. [24] should be conducted in order to compare the therapeutic efficacy of these two compounds with regard to oxaliplatin-associated CIPN.

A closer look into the published reports of the PLIANT [20] and POLAR A and M [19] trials and the prior trial by Coriat suggests that the cited differences are related to an overlooked factor in the study design of the latter [24]. Whereas MnDPDP displayed impressive efficacy, Ca_4_Mn(DPDP)_5_ displayed no efficacy (with regard to CIPN) at any timepoint during ongoing chemotherapy or at follow-up at nine months after the start of chemotherapy. Ca_4_Mn(DPDP)_5_ only showed highly devastating effects. However, in the case of MnDPDP, it was administered directly after the 2 h infusion of oxaliplatin, as a 30 min infusion [24]. In the case of Ca_4_Mn(DPDP)_5_, it was administered as a 5 min infusion starting 10 min before the start of oxaliplatin infusion [19,20]. Considering the complex pharmacokinetics of oxaliplatin as well as that of MnDPDP/Ca_4_Mn(DPDP)_5_, differences in study design may have had dramatic effects on the outcome. Accordingly, the addition of Ca_4_Mn(DPDP)_5_ or MnDPDP ahead of oxaliplatin seems to be a hazardous undertaking. Why the POLAR phase III and the preceding PLIANT phase II trials did not use an identical administration regimen to that used in the preceding MnDPDP trial [24] is therefore difficult to grasp.

Nevertheless, the above-described difference in drug administration between the MnDPDP trial and the Ca_4_Mn(DPDP)_5_ trial further suggests that the negative outcome of the POLAR trials was caused by highly negative interaction between Mn^2+^ [associated with Ca_4_Mn(DPDP)_5_] and Pt^2+^ (associated with oxaliplatin) resulting in a devastating increase in cellular oxidative stress (Figure 2). Taking into consideration that oxidative and nitrosative stress is an unmistakable part of CIPN, including that of oxaliplatin [30,31], points to the interplay between ^•^NO and O_2_^•−^ and generation of strongly oxidizing and nitrating species, connected through the formation of peroxynitrite [32,33] (Figure 3), as the explanation behind the POLAR failure.

A multitude of chemical reactions may lead to the formation of peroxynitrite and tyrosine (Tyr) nitration [32,33], but transition-metal-driven nitration, particularly that of Mn^4+^, is of particular relevance when it comes to platinum-associated CIPN. Furthermore, nitration of Tyr34 in the mitochondrial MnSOD enzyme and Tyr74 in the cytochrome c (Figure 2 and Figure 3), leading to irreversible inactivation of these essential mitochondrial constituents in the DRG, will of course lead to highly devastating effects on the sensory nerve system.

As noted in the Section 1, oxaliplatin and cisplatin might interfere negatively with the cellular handling of copper and further exacerbate the viscous circle of oxidative and nitrosative stress. Interestingly indeed, Coriat, Batteux, et al. [24] analyzed the activity of the SOD enzymes in erythrocyte lysates from participating patients, as described in the material section of [24]. They reported that SOD activity was statistically significantly lower in the non-responder patients than in the responders. Since mammal erythrocytes do not contain mitochondria [34], the reported SOD activity corresponds mainly to the cytosolic CuSOD. The results may indicate a connection between copper status and the severity of Pt^2+^-associated CIPN.

Intriguingly, nitration of Tyr34 in the mitochondrial MnSOD can be catalyzed by intra-enzymatic manganese that has been oxidized into Mn^4+^ [32] (Figure 2). The irreversible inactivation of MnSOD and the disrupted electron transport will in turn severely amplify the initial oxidative insult. Interestingly, CIPN, associated with mitochondrial Tyr34 and Tyr74 nitration, is prevented by the ONOO^−^ decomposition catalyst, MnTE-2-PyP(5+) [30].

## 5. DPDP as a Presumptive Chelation Drug

Both theoretically and experimentally based considerations speak in the direction of another mechanism behind the preventive efficacy of MnDPDP [13,29] instead of the MnSOD-mimetic-based mechanism as suggested by Coriat [24]. An alternative explanation is that DPDP or the metabolites thereof bind Pt^2+^ and act as a chelation drug, i.e., a drug that binds the metal cation in question and facilitates its mobilization and renal excretion (or its redistribution to a non-neuronal compartment) and thereby “grasps the evil by the root” [13]. Into this interplay comes endogenous zinc (Zn^2+^), the most important cation when it comes to in vivo competitive binding to DPDP or its metabolite PLED. Stehr et al., 2019 [13], used the difference in electron paramagnetic resonance (EPR) spectra of MnDPDP and hexaqua-Mn^2+^ to measure the release of Mn^2+^ from DPDP in exchange for Pt^2+^ and Zn^2+^, as described by Schmidt et al., 2002 [35], to obtain an estimate of the formation constant (^10^logK_ML_) of Pt(DPDP) (Figure 4).

EPR-guided competition experiments between MnDPDP with a formation constant (^10^logK_ML_) of about 15 [36] and ZnCl_2_ with a corresponding constant of about 19 [36] or K_2_PtCl_4_ are presented in Figure 4. The resulting competition curve for 100 µM MnDPDP and 10–1000 µM ZnCl_2_ was more or less identical to that presented by Schmidt and co-workers [35]. The corresponding competition curve for 100 µM MnDPDP and 10–1000 µM K_2_PtCl_4_ lay to the left of the ZnCl_2_ curve. The pD_2_ [−^10^log of the concentrations (M) of a drug causing half-maximal responses; EC_50_] (together with 95% confidence interval) values for K_2_PtCl_4_ and ZnCl_2_ were 4.280 (4.227–4.332) and 4.173 (4.127–4.218), respectively; i.e., there was a statistically significant difference between these two pD_2_ values. This suggested that Pt^2+^ in fact has a higher affinity than Zn^2+^ for DPDP. The present curve for Zn^2+^ and that of Schmidt et al. [35] were close to 100% exchange of Mn^2+^ for all concentrations of ZnCl_2_. Importantly, that means the formation constant for PtDPDP may be substantially higher than that for ZnDPDP, but it is not possible to read out how much higher this constant is from the current experiments. The E_max_ (maximal response) values for the K_2_PtCl_4_ and ZnCl_2_ were 95.62 µM (89.54–101.7) and 101.2 µM (95.18–111.2), respectively. There was a clear tendency, although not statistically significant, for the K_2_PtCl_4_ curve to not reach full metal exchange (100 µM). The K_2_PtCl_4_ + MnDPDP samples showed a weak yellow-brownish color during incubation. This was not seen in the MnDPDP control sample or in the ZnCl_2_ + MnDPDP samples, which may suggest that Pt^2+^-driven oxidation of Mn^2+^ had occurred to some extent in the K_2_PtCl_4_ + MnDPDP samples, which in turn may explain the somewhat lower E_max_ than expected.

Whereas Mn^2+^ binds with six coordinates to two phenolates, two amines, and two carboxylates of DPDP or its metabolite PLED, Cu^2+^ binds with four coordinates to the amines and phenolates, with a formation constant of about 22 [36,37]. Because of the smaller ionic radius of Cu^2+^, this metal ion forms a much more stable complex, with a shorter bond distance [38], than that found for the Mn^2+^ ion [36,37]. Similarities between Cu^2+^ and Pt^2+^, hypothetically, suggest that Pt^2+^ may bind to DPDP and PLED in a similar manner to that of Cu^2+^ [13].

## 6. Do DPDP and PLED Fulfill the Necessary Properties of a Platinum Chelation Drug?

A platinum chelation drug, such as DPDP, must fulfill the properties of a selective cytoprotectant, i.e., a drug that protects normal cells but not cancer cells or, in one way or the other, lowers the unwanted toxicity without lowering the tumoricidal efficacy.

MnDPDP has preclinically been shown to protect against unwanted toxicity of oxaliplatin and simultaneously strengthen the tumoricidal efficacy of oxaliplatin in immune-competent balb c mice implanted with CT26 tumor cells [25,26,27]. Similar results have also been reported for Ca_4_Mn(DPDP)_5_ [28,39] (Figure 5). In vitro experiments suggest that the tumoricidal efficacy is mediated by DPDP or PLED and not by their corresponding manganese complexes (Figure 6), hypothetically by an “iron starvation” mechanism [39,40]. These properties of DPDP and PLED are of course promising when it comes to fulfilling the necessary requirements of a selective protectant. The anticancer effects of DPDP and PLED are in themself striking and appear to be worth further study.

Furthermore, copper deficiency induced by the copper chelator tetrathiomolybdate suppresses tumor growth and angiogenesis in an inflammatory breast cancer xenograft in nude mice and Her2/neu cancer-prone transgenic mice [41]. The high affinity of Cu^2+^ for DPDP or PLED may at least theoretically induce copper deficiency and thereby suppress tumor growth, but at the same time, it may induce peripheral neuropathy and hence increase the neuronal insult by adding to the effect of oxaliplatin. The finding that the platinum- and copper-binding chelator STS significantly impairs the tumoricidal efficacy of cisplatin in pediatric patients with disseminated solid tumors but not in patients with localized, non-metastatic solid tumors [14,15] apparently illustrates the difficulties in predicting the tumoricidal effect of copper deficiency. Whereas the signaling networks that integrate fluctuations in the abundance of growth factors, nutrients, and metabolites are well established, the discovery of signaling molecules capable of mediating similar functions depending on copper availability is rudimentary [42].

Crucial for achieving the goal of a selective platinum chelation drug are the pharmacokinetic and pharmacodynamic properties of the actual platinum drug, in this case, oxaliplatin. The very high distribution volume of Pt^2+^ after intravenous administration of oxaliplatin, exceeding 300 liters [43], is of immense importance. That is a property promoting rapid intracellular uptake and cellular retention. The high distribution volume of Pt^2+^ is apparently due to the lipophilic character of the tumoricidal active metabolite, platinum(II)dichloro-(trans-l-1,2-diaminocyclohexan) [Pt(DACH)Cl_2_], and the subsequent binding of platinum to proteins, DNA, and other cellular constituents [43].

Oxaliplatin undergoes extensive non-enzymatic biotransformation in plasma ultrafiltrate and urine, and at least seventeen Pt^2+^-containing metabolites are observed after 24 h [43]. A crucial point is of course whether these metabolites contribute to the tumoricidal efficacy of oxaliplatin. A retrospective comparison by McWhinney et al., 2009 [44], of the neurotoxicity versus the response rate for platinum drug treatment from fourteen Pt^2+^-containing trials, including oxaliplatin, cisplatin, and carboplatin, did not indicate any association between neurotoxicity and tumoricidal efficacy. McWhinney et al. concluded that neurotoxicity is not merely a ‘‘necessary evil’’ but can be approached as an avoidable side effect of a platinum agent.

The Pt(DACH)Cl_2_ in the plasma ultrafiltrate from 10 patients receiving oxaliplatin was determined using high-performance liquid chromatography and atomic absorption spectrometry [45]. Less than 3% was found undergoing biotransformation into the cell-permeable and active Pt(DACH)-Cl_2_ metabolite. After uptake into a tumor cell (or a normal cell), either by diffusion or active transport and due to the low intracellular concentration of chloride, Pt(DACH)-Cl_2_ will undergo sequential hydrolysis of the chlorides, which will enable crosslinking at guanine residues at N7 [17]. Key elements in the tumoricidal activity of oxaliplatin or cisplatin are hence (i) intracellular hydrolysis, (ii) binding to guanine bases, (iii) distortion of DNA, and (iv) changing its interactions with proteins, leading to cell killing by apoptosis.

Most platinum anticancer complexes have the general formula cis-[PtX_2_(NHR_2_)_2_], in which R is an organic fragment and X is a leaving group, such as chloride in cisplatin or a chelating carboxylate in oxaliplatin. Typically, platinum coordination compounds have thermodynamic strength of 100 kJ/mol or below, much weaker than covalent, C–C and C–N or C–O single and double bonds, the strength of which is between 250 and 500 kJ/mol [17]. However, the ligand-exchange behavior of Pt compounds is quite slow, which gives them a high kinetic stability and results in ligand-exchange reactions of minutes to days, rather than microseconds to seconds for many other coordination compounds. Furthermore, Pt^2+^ has a strong thermodynamic preference for binding to S-donor ligands, and with so many cellular platinophiles, i.e., S-donor ligands, such as glutathione and methionine, as competing ligands in the cytosol, it appears highly relevant to ask whether the few percent of PtDACH-Cl_2_ formed will ever reach DNA [17].

The answer to the question seems to be the migration of Pt^2+^ from an S-ligand to purine N7, according to Reedijk 2003 [17]. The same arguments may also be valid for the two amines in DPDP and PLED, involved in the binding of Cu^2+^ [13], and hypothetically in the binding of Pt^2+^ in competition with the dominating non-tumoricidal Pt^2+^ ligands. Stoichiometrically, the 5 µmol/kg MnDPDP dose used in the Coriat study [24] is equivalent to the standard dose of 85 mg/m^2^ oxaliplatin in a 90 kg patient with a body area of 2 m^2^. As the dose-limiting factor in the use of MnDPDP is neuronal retention of manganese and in case DPDP chelation therapy is shown to be clinically feasible, there is apparently room for increasing the DPDP dose.

We conclude that the devastating worsening of oxaliplatin-associated CIPN in the POLAR trials, as presented by Pfeiffer et al., 2022 [19], was presumably due to co-retention of Ca_4_Mn(DPDP)-associated Mn^2+^ and oxaliplatin-associated Pt^2+^ in DRG neurons, causing oxidative and nitrosative stress, site-specific Tyr nitration, and irreversible inactivation of the mitochondrial MnSOD enzyme and cytochrome c. These are extremely devastating events that lead to serious injuries in the peripheral sensory nerve system. If Pt^2+^ binds to DPDP with high enough affinity in vivo as previously demonstrated under in vitro conditions by Stehr et al. [13], the use of manganese-free DPDP may in fact solve the problem with oxaliplatin-associated CIPN. Furthermore, using a drug administration design identical to that used by Coriat et al. in 2014 might potentially enable the safe use of MnDPDP [and maybe also Ca_4_Mn(DPDP)_5_] to prevent oxaliplatin-associated CIPN, in case DPDP fails. The most important lesson to be learned from the POLAR trials is the danger of proceeding from phase II into extensive phase III trials in an ad hoc manner, without supportive phase II data.

## Figures and Tables

**Figure 1 ijms-25-04347-f001:**
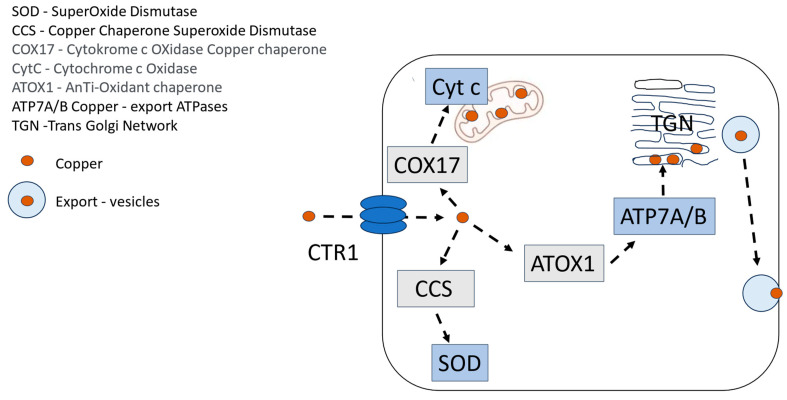
Schematic illustration of the tightly regulated cellular handling of copper, where CuSOD (SOD) and cytochrome c (Cyt c) are of particular importance for cells with high energy demand, such as dorsal root ganglion (DRG) cells. Copper Transporter 1 (CTR 1) is implicated in the uptake of oxaliplatin and other Pt^2+^-containing drugs.

**Figure 2 ijms-25-04347-f002:**
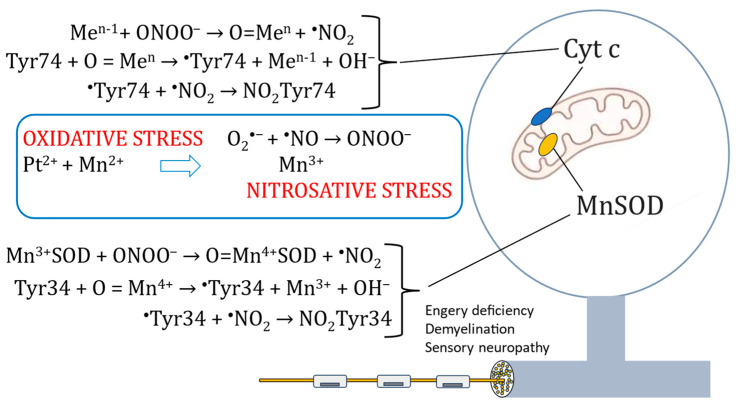
Schematic illustration of a DRG and two important targets for Pt^2+^-associated oxidative and nitrosative stress, namely mitochondrial MnSOD and cytochrome c (Cyt c). The transition metal (Me)-driven chemistry will lead to site-specific nitration of Tyr34 of the MnSOD-enzyme and Tyr74 of cytochrome c, irreversible inactivation of MnSOD, and disruption of normal electron transfer in the respiratory chain. In the upper reaction scheme, the endogenous transition metal (Me) could be Mn, Fe, or Cu.

**Figure 3 ijms-25-04347-f003:**
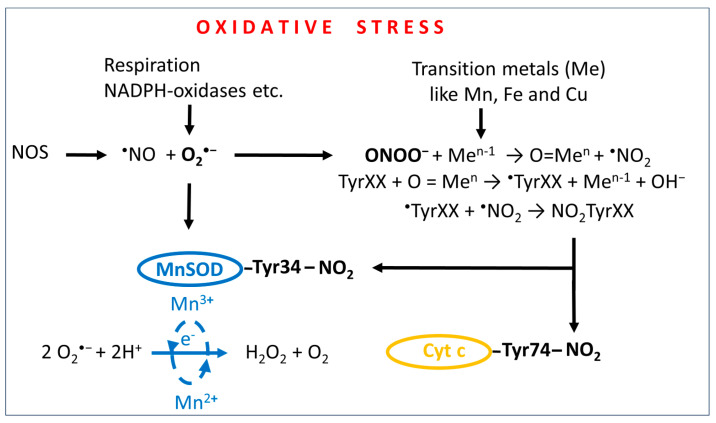
Schematic illustration of the connection between oxidative and nitrosative stress. Under normal redox conditions, MnSOD (together with CuSODs) will keep the production of superoxide (O_2_^•−^) in check by dismutation of O_2_^•−^ into hydrogen peroxide (H_2_O_2_) and water. Under severe oxidative stress, however, the production of O_2_^•−^ will exceed the capacity of the SOD enzymes, and O_2_^•−^ will react with ^•^NO and form highly toxic peroxynitrite (ONOO^−^). Driven by the reduction of Mn^4+^ to Mn^3+^, ONOO^−^ will nitrate tyrosine residues Tyr34 and Tyr74, which will cause mitochondrial MnSOD inactivation and disruption of electron transport through the respiratory chain. These are highly devastating effects on high-energy-demand tissue, such as that of the peripheral sensory nerve system. Me is a transition metal (Mn, Fe, or Cu).

**Figure 4 ijms-25-04347-f004:**
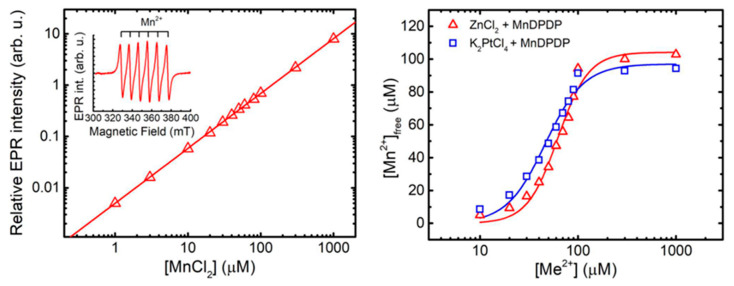
The EPR standard curve for free Mn^2+^ is shown in the left graph. The double integral of the EPR signal (insert) in arbitrary units is plotted against the concentration of MnCl_2_. Metal exchange of MnDPDP by Zn^2+^ and Pt^2+^ is shown in the right graphs. The EPR invisibility of MnDPDP was used to monitor the exchange of Mn^2+^ for either Zn^2+^ or Pt^2+^ (Me) [13].

**Figure 5 ijms-25-04347-f005:**
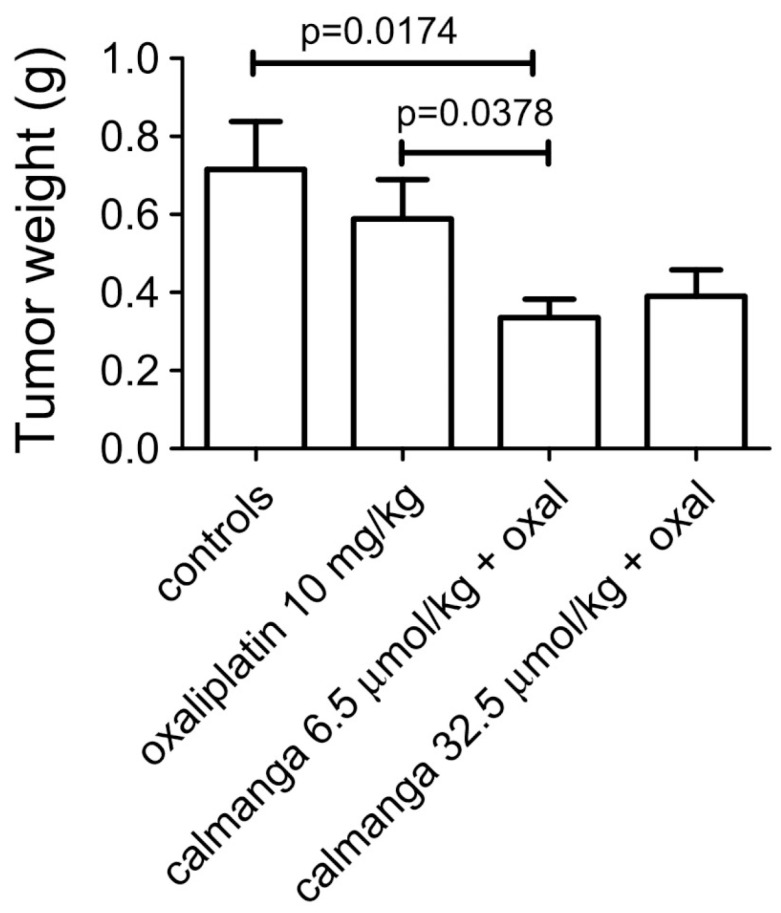
Antitumor effect of oxaliplatin (10 mg/kg) in the absence and presence of 6.5 and 32.5 µmol/kg calmangafodipir [Ca_4_Mn(DPDP)_5_] in immune-competent balb c mice implanted with CT26 tumors. Results are expressed as mean ± SEM (*n* = 5) [28].

**Figure 6 ijms-25-04347-f006:**
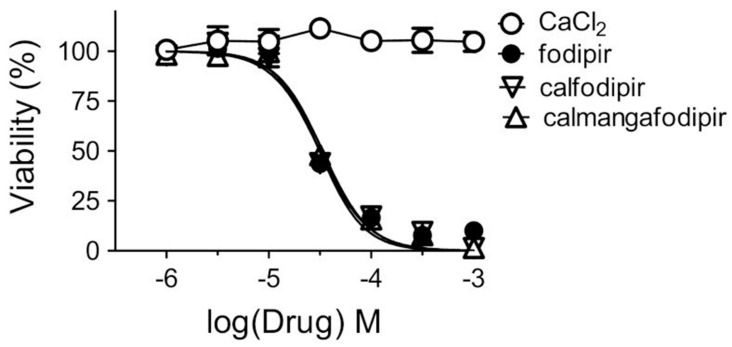
Cytotoxic effects of increasing concentrations of CaCl_2_, fodipir (DPDP), calfodipir (CaDPDP), and calmangafodipir [Ca_4_Mn(DPDP)_5_] in CT26 tumor cells (mean ± SD; *n* = 3) [28].

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
