# Peer review of "Manganese- and Platinum-Driven Oxidative and Nitrosative Stress in Oxaliplatin-Associated CIPN with Special Reference to Ca4Mn(DPDP)5, MnDPDP and DPDP"

_ijms, 2024, doi:10.3390/ijms25084347_

Round 1

Reviewer 1 Report

Comments and Suggestions for Authors

The authors describe the interaction between Mn compounds containing the DPDP ligand with the Pt(II) center present in the antitumor oxaliplatin. The redox interaction between both (Pt(II) and Mn(II)) generates oxidation of Mn(II) to Mn(III) and Mn(IV) and depletion of MnSOD, with resulting destruction of the Mn(II) complex containing DPDP . Depletion of MnSOD results in an increase in the levels of superoxide radical, which reacts with nitric oxide, forming ONOO-, generating redox imbalance and nitrosative stress.

The motivation for this publication is strongly based on literature. I suggest including the drawing of the Mn compounds and the DPDP ligand. Generally speaking, DPDP acts as a chelator that interacts with Pt(II). It was not clear what proportion of each Mn compound and oxaliplatin was used. I suggest monitoring the UV_Vis reaction of the DPDP exit from the Mn(II) compound using spectroscopy, at physiological pH or saline. I also suggest cell cultivation of some cell line sensitive to oxyliplatin, to analyze the reduction of the antitumor efficiency of the drug in the presence of Mn compounds, thus, the authors' proposal that the interaction between DPDP and Pt(II) would reduce the antitumor activity of oxyplatin and consequently the CIPN must be validate.

After the coordination of DPDP to the Pt(II) center, there is the rupture of the complexes. So, a question arises: would it be possible to synthesize a Pt(II) compound with DPDP and investigate its antitumor activity? It could be characterized by NMR. It could also test the reaction between DPDP and Zn(II), since its depletion is associated with immune system malfunction. If it is inactive, it would be another point in favor of the proposal, which indicates that the reduction in neuropathy is due to the inactivation of the antitumor itself.

I would like the ask the authors to provide more data on the interaction between the Mn(II) and Pt(II) complexes. Figure 3 gives the impression of  cations and not complexes, and free metal could favor oxidative stress. Would Mn(IV)SOD be stable? In Fig 4 the authors mention that MnSOD cycles between Mn(II) and Mn(III). FIg 3 highlights the nitrosylation of tyrosine, and its inactivation can lead to deleterious effects.

I agree with the authors that phase II studies should be carried out more carefully to support phase III studies.

In Figure 2, details such as solvent (saline, water) are missing so that we can justify the coordination of the chlorine and water ligands. When it enters the cell, the chloride concentration drops and coordinates with water, similar to cisplatin. I suggest you add these details.

Author Response

The authors appreciate the overall positive comments from Reviewer 1.

When it comes to the relevant suggestions of additional experiments, the authors want to stress stress that the current article is a short review/commentary type of article discussing already published results. These suggestion could therefore not be addressed in the current manuscript. 

Reviewer 2 Report

Comments and Suggestions for Authors

This is a somewhat interesting review and hypothesis paper on the results of clinical trials involving the use of potential antioxidants to prevent toxic side effects of platinum-based anticancer drugs, although the hypothesis is presented that the agents shopwing beneficial effects might actually be acting as chelating ligands for Pt.  The manuscript is reasonably well written and timely

The manuscript contains numerous gtrammatical errors and sloppy formatting errors.  The first paragraph of the abstract is not fully justified.  The caption for Figure 8 is on a different page than Figure 8.  In the references, the second line of reference 37 is numbered as reference 38, so that all subsequent reference numbers are wrong.

The figures need assistance in many cases.  In figure 1, the small orange circles are not defined as copper; the larger light blue circles are not definede (and I do not know what they are suppose to be).  In figure 2 for the final product, the top base needs to be rotated so that the Pt-N bond is reasonable.  On the right side of Figure 5, the x-axis is labelled "[Me2+]", but Me is not defined or is supposed to be Mn2+???  For fighure 6, the lower left oxygen atom that is bound to the Pt is out of place; the platinum does not appear planar.  In figure 7, the x-axis should read "log[Drug] (M)".

Comments on the Quality of English Language

Numerous sentence fragments are present, along with numerous grammatical errors of a more technical nature.  Requires proofiung by an English language expert.

Author Response

The authors appreciate that Reviewer 2 finds the current manuscript interesting and reasonably well written. 
Errors have been corrected, including the reference list and Figures 1, 5, and 6. In addition. Figures 3 and 4 have been revised, by changing “M” to “Me” to fit in with Figure 5. Figure 4 have been revised to better fit in with Figure 4.     

Round 2

Reviewer 2 Report

Comments and Suggestions for Authors

Some items with the figures were not addressed.

In figure 1, the larger light blue circles carrying copper to the membrane are still not defined  In figure 2 for the final product, the top base needs to be rotated so that the Pt-N bond is reasonable.  For figure 6, the lower left oxygen atom that is bound to the Pt is still out of place; it is not bound to the rest of the organic molecule.  The compound is assuredly square planar so deleting "square planar" from the caption is not helpful.

Author Response

Figure 1 has been revised in accordance with the comments from the reviewer.

Figures 2 and 6 were included with the purpose to facilitate for the readers. As pharmacologists, the authors have neither the capability nor the capacity to construct perfect molecular drawings. The authors have therefore decided to withdraw Figure 1 and 6 from the manuscript, and relevant changes have been done in the running text. 

Changes marked in red colour .
